# Development of Single-Molecule Electrical Identification Method for Cyclic Adenosine Monophosphate Signaling Pathway

**DOI:** 10.3390/nano11030784

**Published:** 2021-03-19

**Authors:** Yuki Komoto, Takahito Ohshiro, Masateru Taniguchi

**Affiliations:** 1Institute of Science and Industrial Research, Osaka University, 8-1 Mihogaoka, Ibaraki, Osaka 567-0047, Japan; komoto@sanken.osaka-u.ac.jp (Y.K.); toshiro@sanken.osaka-u.ac.jp (T.O.); 2Artificial Intelligence Research Center, The Institute of Scientific and Industrial Research, Osaka University, 8-1 Mihogaoka, Ibaraki, Osaka 567-0047, Japan

**Keywords:** DNA, cyclic AMP, single-molecule detection, nanogap, second messenger

## Abstract

Cyclic adenosine monophosphate (cAMP) is an important research target because it activates protein kinases, and its signaling pathway regulates the passage of ions and molecules inside a cell. To detect the chemical reactions related to the cAMP intracellular signaling pathway, cAMP, adenosine triphosphate (ATP), adenosine monophosphate (AMP), and adenosine diphosphate (ADP) should be selectively detected. This study utilized single-molecule quantum measurements of these adenosine family molecules to detect their individual electrical conductance using nanogap devices. As a result, cAMP was electrically detected at the single molecular level, and its signal was successfully discriminated from those of ATP, AMP, and ADP using the developed machine learning method. The discrimination accuracies of a single cAMP signal from AMP, ADP, and ATP were found to be 0.82, 0.70, and 0.72, respectively. These values indicated a 99.9% accuracy when detecting more than ten signals. Based on an analysis of the feature values used for the machine learning analysis, it is suggested that this discrimination was due to the structural difference between the ribose of the phosphate site of cAMP and those of ATP, ADP, and AMP. This method will be of assistance in detecting and understanding the intercellular signaling pathways for small molecular second messengers.

## 1. Introduction

Second messengers are important for the signal transduction of intracellular signals because they are always released by the cell in response to exposure to extracellular signaling molecules and trigger physiological changes at the cellular level, such as proliferation, differentiation, migration, survival, apoptosis, and depolarization [1,2,3]. Therefore, measuring the molecular behavior of second messengers is important to understand the signal transduction events that occur inside cells. Among these second messengers, cyclic adenosine monophosphate (cAMP) is an important research target because it activates protein kinases, which are used to regulate the passage of ions and small molecules such as Ca^2+^ through ion channels [4,5,6]. This cAMP is produced from adenosine triphosphate (ATP) by a g-protein enzyme and is converted to ATP by an adenylate cyclase enzyme. As a result that the chemical properties and structures of nucleic acid molecules (i.e., ATP, adenosine monophosphate (AMP), and adenosine diphosphate (ADP)) are very similar to those of cAMP, the cAMP should be selectively detected in order to detect the chemical reactions related to its intracellular signaling pathway.

Fluorescence optical imaging using optically synthesized molecular or protein probes has been as a selective detection method for cAMP. These probes are composed of two functional parts that selectively bind to cAMP and have the ability to detect a fluorescent or luminescent signal inside cells [7,8,9,10]. However, in order to simultaneously detect multiple targets consisting of cAMP-related molecules such as ATP, cAMP, and ADP to understand the signal transduction of the cAMP pathway, each target requires a separate probe. In addition, because these probes are bound to the individual target molecules in the recognition, the detection conditions are somewhat different from the physiological conditions, and it is appropriate to directly measure each of these targets. Recently, single-molecule detection nanotechnologies, such as nanopore and nanogap devices, have developed so that these methods become promising approaches, which provide their potential applications within the life science fields [11,12,13,14,15,16]. Among them, one of the approaches is single-molecule quantum detection, which is based on the following detection principle. When sample molecules pass between nanogap electrodes that are used as sensors, a tunnel phenomenon induces an electron transfer through the molecules, resulting in the detection of the individual electrical conductivity due to the electronic state of each of the molecules [14,15,16,17]. To date, we have reported the identification of various biomolecules such as DNA, RNA, and amino acids using single-molecule quantum detection methods [18,19,20,21,22].

The current study investigated cAMP and its related molecules using the single-molecule quantum detection method (Figure 1a), and succeeded in electrically detecting cAMP at the single-molecule level. The developed machine learning method was also successfully used to discriminate the cAMP signals from those of ATP, AMP, and ADP (Figure 1b), which are molecules similar to cAMP and coexist under physiological conditions. The single cAMP signal discrimination accuracies from AMP, ADP, and ATP were found to be 0.82, 0.70, and 0.72, respectively. These values indicated a 99.9% accuracy when detecting more than 10 signals. Furthermore, it was found that the features that significantly contributed to the discrimination by the developed machine learning method were affected by the interaction of the electrodes and target molecules. This result indicated that the structural difference from the ribose of the phosphate site of cAMP enabled it to be distinguished from ATP, ADP, and AMP. Thus, it can be said that the single-molecule quantum measurement method has shown the potential for identifying one molecule of cAMP and visualizing the information transmission system in the body containing the cAMP.

## 2. Materials and Methods

### 2.1. Single-Molecule Electrical Measurement of Sample Nucleotide Solution

5′-Adenylic acid (AMP), 3′,5′-cyclic adenosine monophosphate (cAMP), adenosine 5′-trihydrogen diphosphate (ADP), and adenosine 5′-tetrahydrogen triphosphate (ATP) were purchased from Sigma-Aldrich (St. Louis, MO, USA) and used without further purification. Their chemical structures are shown in Figure 1b. These were used to prepare 10 µM aqueous solutions without further purification. The prepared sample solutions were dropped (20 µL for each measurement) at the center of a sensor plate that contained the fabricated nanogap electrodes.

The nanogap electrodes were constructed using the mechanically controllable break junction (MCBJ) nanofabrication method (Figure 1c). The procedures for fabricating MCBJs are detailed elsewhere [23,24,25]. In this study, a nanogap device was fabricated as follows. First, a silicon substrate was electrically insulated by applying a thin polyimide layer. A nano-gold junction was fabricated using electron-beam lithography on the substrate. Next, a SiO_2_ layer was deposited using chemical vapor deposition. Here, a nanochannel pattern was superimposed on the nano-gold junctions by electron beam lithography. Finally, this pattern was developed, and dry etching was performed to form nanochannels. A cover made of polydimethylsiloxane (PDMS) was attached to the silicon substrate. The PDMS cover had a microchannel that connected the hole for introducing the sample solution and the nanochannel of the sensor in advance. The PDMS was purchased from Toray Dow Corning. Finally, the PDMS cover and silicon substrate were treated with ozone plasma and then bonded.

The nanogap electrode devices were used for single-molecule measurements of the sample nucleotide solutions. The gap size was set to 0.65 nm and finely tuned by the piezoelectric element during all the measurements. During the measurement, the gap distance was maintained by feedback control of the piezo actuators. The gap distance was estimated using the following equation for the direct tunneling current:(1)$$I=const\exp\left(-\frac{4\pi }{h}\sqrt{2mwl}\right)$$
where *h*, *m*, *w*, and *l* represent the Planck constant, electron mass, work function of the gold electrode, and gap distance, respectively. An electron mass of 9.1 × 10^−31^ kg was used for m, and the work function of Au(111) (5.1 eV) was used for w [19].

After every 1 h period of current–time *(I*–*t*) measurements, the MCBJ sample was replaced with a new one. The current across the electrodes was amplified by a custom-built logarithmic current amplifier and recorded at 10 kHz and 100 kHz using an NI PXIe-4081 digital multimeter (National Instruments, Austin, TX, USA) and NI PXI-5922 (National Instruments, Austin, TX, USA) under a DC bias voltage of 100 mV. More than three sets of measurements were performed using different gold-gap sensors.

### 2.2. Single-Pickup and Machine Learning Method for Single Analysis

The signal-rise time was judged to be the start of the signal from the estimated baseline with a value exceeding six times the “noise level” as the threshold value. At this time, the signal exceeded the noise level. Next, the signal was determined to have ended when a data point below the noise level appeared from the estimated baseline after the signal started. The baseline was defined as the mode of the histogram of the last 2000 data points at each time. The noise level was defined as the mode of the standard deviation of 200 data points obtained by dividing the latest 10,000 data points at each time into 500 equal parts.

A machine learning analysis was performed using Python 3.6. First, positive and unlabeled data classification (PUC) was performed to remove “blank signals“, which were also observed in blank solutions because of the migration of gold atoms from the electrode and contamination and reported in previous studies [26,27,28,29,30]. The PUC was an appropriate algorithm for blank signal removal. The PUC algorithm is based on the method of Elkan and Noto [31]. The inner classifier was the Gaussian naïve Bayes from the scikit-learn library, version 0.21.3 [32]. After blank signal removal, the signals were trained and classified using supervised machine learning with the random forest classifier from the scikit-learn library, version 0.21.3 [32]. A 10-fold cross-validation was performed, where all the data of the single-molecule signals were partitioned into 10 sub-datasets, and 10 classifications were performed for each of the sub-datasets used as test data, with other sub-datasets used as training data. The average ratios of the 10 classifications are shown in the confusion matrices.

## 3. Results

### 3.1. Single-Molecule Electrical Detection of cAMP, ATP, AMP, and ADP

First, the cAMP solution was investigated using the nanogap electrodes (Figure 1a: schematics). The current–time (*I–t*) profiles for the blank and cAMP solutions are shown in Figure 2a,b, respectively. Further current-time traces of cAMP measurements are shown in Appendix A. The cAMP molecules were detected as characteristic pulse current signals because increases in the tunnel current were induced by the cAMP molecules translocating through the nanogap electrodes. A characteristic current signal was detected, as shown in Figure 2c. The signal’s peak value was defined as *I_p_*, and the signal duration time was defined as t_d_. Compared to the cAMP signals in Figure 2b, the average *I_p_* for the characteristic current signals decreased from 33 pA (Figure 2b) to 18 pA (Figure 2a). In addition, the signal frequency decreased significantly (Figure 2d). The duration-time was found to be 0.3 milliseconds. In the previous study, it was found that the signal intensity was around 10 pA, and the duration time was around 0.2 milliseconds. Such statistical signal values such as signal intensities and duration-time were very similar to those of the blank signals obtained in buffer aqueous solutions in previous studies [19]. These results suggested that the obtained current signals were typical molecular signals for cAMP molecules.

Next, single-molecule measurements were made of the ATP, AMP, and ADP. To evaluate the differences among these four types of molecular signals, the peak current (*I_p_*) values were statistically compared. Figure 3a–d show the current signal histograms. The average *I_p_* values of cAMP, AMP, ADP, and ATP at a bias of 100 mV were 32 ± 18, 27 ± 13, 41 ± 21, and 36 ± 15 pA, respectively. These results demonstrated that the differences in the conductance values were small. Similarly, the duration (*t_d_*) values were statistically compared. Figure 3e–h show the signal duration time histograms. The average signal duration-time were 4.6, 4.6, 4.1, and 5.3 ms for cAMP, AMP, ADP, and ATP, respectively. These duration time values (*t_d_*) were also similar, although the ATP signals had slightly longer averaged t_d_ values. Thus, statistical analyses of these signals showed that a single feature parameter (i.e., *I_p_* or *t_d_*) would not be sufficient for their discrimination. However, multiple feature parameter analysis has potential for improving the discrimination accuracy of cAMP, ATP, AMP, and ADP. Figure 3i–l show two-dimensional plots of *I_p_* and *t_d_*, and the dependency of the sample species appears in these plots. The histograms of cAMP and AMP are similar. But the difference was observed as shown in Appendix A in Supplymentary Material. These results indicated that information about the entire waveforms of the molecular signals or several characteristic feature parameters extracted from the signals are needed for accurate discrimination.

### 3.2. Discrimination of cAMP ATP, AMP, and ADP Using Machine Learning Method

To improve the discrimination of cAMP, ATP, AMP, and ADP using a multiple feature parameter analysis, a machine learning-based classification method was applied, with multidimensional feature parameters prepared using each of the individual signals. The scheme of this machine learning method is shown in Figure 4a,b. In the first step, each signal was obtained by the single-molecule electrical measurement. In the second step, the single-molecule signal was extracted based on the criteria described in the materials and methods section. This study used the following 13 feature parameters: the peak value of the signal current (*I_p_*), average signal value (*I_ave_*), duration time (*t_d_*), and ten-dimensional shape factors (*S_n_ = I_n_/I_p_* (*n =* 1, 2, …, 10)). The values of *S_n_* were defined as follows. The current–time (*I–t*) profile of a signal region was separated into ten time regions, and then each of the average current values (blue circles) for each of the time regions (*I*_1_, *I*_2_, *I*_3_, *I*_4_, *I*_5_, *I*_6_, *I*_7_, *I*_8_, *I*_9_, and *I*_10_) was calculated (Figure 4c). Each of these current values was normalized by the maximum current value of the signal (*I_p_*), and the values in each region were defined as the shape factors, i.e., *S*_1_, *S*_2_, *S*_3_, *S*_4_, *S*_5_, *S*_6_, *S*_7_, *S*_8_, *S*_9_, and *S*_10_. In the third step, each of the extracted signals was converted into feature parameter data (*I_p_*, *I_ave_*, *t_d_*, *S_n_ (n =* 1, 2, …, 10)). The dataset of the feature parameter data was split into two datasets: the training data and test data. In the fourth step, a machine learning classifier learned the feature parameters from the training data, as shown in Figure 4b. In the fifth step, the classifier was utilized to identify each of the chemical species in the test data signals for cAMP, ATP, AMP, and ADP, and the discrimination accuracies were evaluated as F-measure values. In the final step, the accuracy was validated by performing a 10-fold cross-validation, as described in the materials and methods section.

These feature parameters (*I_p_*, *I_ave_*, *t_d_*, and *S_n_*) were used to perform classifications with the supervised machine learning method for all the different discrimination combinations of cAMP, ADP, AMP, and ATP. The evaluation results are shown in Figure 4d. The discrimination accuracy value of 0.28 for cAMP was poor, while those for the ADP, AMP, and ATP signals showed high accuracy.

In the single-molecule detection, the obtained signals include target sample signals and also include signals which were generated from the fluctuation of the molecule or the electrode as shown in Figure 2a,d. We called these signals blank signals. The blank signals are not Johnson noise, (thermal noise), and/or shot noise of measuring instruments generated by electrical measurement, but the signal. In the previous reports, this blank signal adversely affected the discrimination of molecules [26,27,29] so that the reduction of blank signal are a key step for the discrimination with high accuracy. To address the issues of the blank signals, the signal of the sample target, i.e., cAMP, ADP, AMP, and ATP, had to be extracted from the obtained signals.

In order to remove the blank signals, the PUC method was applied to this signal discrimination. In the PUC method, the signals obtained from the blank solution (control experiments) were used as “positive” signals. On the other hand, the signal data of the sample solution were regarded as “unlabeled” data because the signals obtained from the sample solution contained both blank signals and the sample molecule signals. The scheme is shown in Figure 5a,b. In the first step, single-molecule measurements were performed for the solvent (blank). The signal data of the blank were regarded as positive data. In the second step, each signal was converted into a feature parameter value (*I_p_*, *I_ave_*, *t_d_*, and *S_n_*). In the third step, the machine-learning classifier learned the feature parameters of the blank signals. In the fourth step, single-molecule measurements were performed on the sample solutions, and these signal data were regarded as unlabeled data. In the fifth step, the sample molecule signals were discriminated from the blank signals using the machine learning classifier of blank signals learned in the third step. In the final step, the extracted sample data (cAMP, ADP, AMP, and ATP) were used for the subsequent classification by the supervised machine learning method.

Figure 5c shows the evaluation results for the cAMP, ADP, AMP, and ATP discrimination accuracies using the PUC method. Compared to the discrimination accuracy value of 0.25 for cAMP without the PUC method (Figure 4d), the discrimination accuracy of cAMP was significantly improved to 0.44. Together with the other discrimination accuracies of 0.50 for AMP, 0.49 for ADP, and 0.59 for ATP, the average discrimination accuracy was also improved to 0.50. For instance, when more than ten signals with a discrimination accuracy of 0.50 were obtained, a 99% accuracy is obtained (Figure 5d). In the case of identification with only two types, the discrimination accuracy values were found to be 0.70 for AMP/cAMP, 0.69 for ADP/cAMP, 0.75 for ADP/cAMP, 0.69 for AMP/ATP, 0.74 for ATP/cAMP, and 0.69 for ADP/ATP (Figure 5e–j). The obtained accuracy represents that the specific molecular species were identified with a certain accuracy for each signal. Therefore, in order to determine the ratio of sample molecules in the mixed solution, the discrimination error of each signal can be determined from the obtained confusion matrix such as Figure 5c. In the previous report, we discriminated ethylated guanosine from guanosine and estimated the mixture ratio [27], and also identified each sample in a mixed solution containing dopamine, serotonin, and noradrenaline and estimated the mixture ratio, which is comparable to that of the prepared solution [26]. Therefore, this method will be applicable for understanding cAMP-related reactions such as enzymatic reactions at the single-molecular level. However, we also found the determined mixture ratio has some difference which depends on the sample species. Since the discrimination errors were due to the difference between the concentration of bulk solution and that of near the sensor gap, further development of the active mass transport function in solution such as electrophoresis [30] would be needed.

## 4. Discussion

This molecular recognition method using machine learning (Figure 5a,b) had at least two advantages compared to the previous discrimination method that only analyzed a single parameter such as *I_p_* or *t_d_*. The first advantage was the reduction of the blank signal using the PUC method before signal discrimination. The discrimination accuracy was significantly improved after PUC (Figure 5c) compared to that before PUC (Figure 4d). In addition, after PUC, the average *I_p_* values of the cAMP, AMP, ADP, and ATP signals were 120 ± 18 pA, 100 ± 24 pA, 104 ± 13 pA, and 148 ± 22 pA, respectively. As a result that the intensity of blank signals is known to be small, the difference in the currents with and without PUC was due to the blank signal reduction when using the PUC method. Therefore, the PUC method allowed the target signals to be extracted from the obtained signals, which made it possible to better detect slight differences in the chemical structures compared to conventional methods.

The second advantage was that by using multiple parameters (*I_p_*, *I_ave_*, *t_d_*, *S_n_* (*n =* 1, 2, …, 10)) for each of the signals, this analysis significantly improved the discrimination accuracy and understanding of the differences in the physical behaviors of each molecule. In this analysis, the maximum current value (*I_p_*), average value (*I_ave_*), signal duration (*t_d_*), and signal shape factor of each region (*S_n_*) were used for supervised machine learning. A comparison of several classification algorithms, including support vector machine (SVM) and extreme gradient boosting (XGBoost), showed that the random forest algorithm provided the best discrimination accuracy (Figure 6a). In an ensemble learning method such as random forest, some feature parameters are randomly selected for classification trees, and then classification is performed using the ensemble. If only a specific feature parameter is important for discrimination, while other parameters are not important, the identification accuracy in a random forest becomes low. Therefore, the good discrimination accuracy using the random forest method indicated that all the prepared features used here were involved in the discrimination.

Next, to evaluate each feature parameter’s contribution to the discrimination, the discrimination accuracy was determined after changing the combination of feature parameters used. Among the four sets of parameter combinations, the best combination was found to be the data sets of the maximum current value (*I_p_*), average current value (*I_ave_*), t_d_, and shape factors (*S_n_*). This confirmed that all of the prepared features used here were useful for the discrimination. Importantly, among the three types of features, (*I_p_*, *I_ave_*), (*I_p_*, *I_ave_*, *t_d_*), and the shape factors (*S_n_*), the shape factors were found to have the highest discrimination accuracy of 0.44, compared to 0.40 for the maximum and average values of the current and the signal duration (*I_p_*, *I_ave_*, *t_d_*), and 0.39 for the average and maximum values of the current (*I_p_*, *I_ave_*) (Figure 6b). It is suggested that the shape factors were important because the translocation behaviors inside the nanogap electrodes differed depending on the size and polarization of the molecule, resulting in characteristic signal behaviors.

Finally, the importance of each feature parameter was estimated to quantitatively investigate which feature was more effective for discrimination. The random forest classifier could evaluate the importance of each feature [33], which was an index of how much the discrimination accuracy decreased when that feature parameter was omitted. Figure 6c shows the importance index for each feature parameter. Among these indexes, the two most important feature parameters were found to be the maximum current value (*I_p_*) and first shape factor (*S*_1_).

For the maximum current value (*I_p_*), an increase in the signal current was induced by electron transmission via a single molecule at the nanogap electrodes. It is well known that the current intensity is proportional to the transmission of a single-molecule junction, *τ*, as represented in the following equation [34,35,36].
(2)$$\tau =\frac{4\Gamma_{L}\Gamma_{R}}{\varepsilon^{2}+{\left(\Gamma_{L}+\Gamma_{R}\right)}^{2}}$$

Here, *ε* is the conductive molecular orbital alignment from the gold Fermi level, and Γ*_L_* and Γ*_R_* denote the coupling values between the molecule and the left and right electrodes, respectively. A change in the molecular orbital position causes a change in the single-molecule conductance. It has been previously reported that the difference in conductivity is closely related to the highest occupied molecular orbital (HOMO) level [15,16,19,26,37,38]. In this study, all of the molecules investigated (cAMP, ATP, ADP, and AMP) have an adenine structure, which is the main local electron density site and related to the HOMO level of the molecule. The electron density of the adenine structure is influenced by the position of the phosphate group, which is an electron-withdrawing functional group. For instance, the phosphate group of cAMP approaches the vicinity of adenine because of two covalent bonds between the phosphate and ribose. Thus, differences in the sizes of the molecules reflect those at each of the HOMO levels, resulting in signal intensity differences.

The electronic structure of molecules, such as HOMO levels, is one of the important parameters for each of the signal characters. In addition to the electronic structure of molecules, it was also found that fluctuations of sample molecules inside electrodes, i.e., structural motion of the DNA and its environment, are closely related to the signal characters [15,16]. In this study, among various potential candidates, we utilized *t_d_*, *S_n_* (*n* = 1, 2, …, 10)) for this discrimination as feature parameters for this machine learning because the characteristic fluctuation, which should be influenced by the molecular orientation and interaction inside nanogap, would induce the characteristic signal shape and signal duration time. Among the shape factors of *S_n_* (*n* = 1, 2, …, 10)), the first shape factor (S_1_) was the first region of the ten separated regions (Figure 4c), i.e., the initial rising current for each signal. The difference in *S*_1_ was due to the initial interaction between the molecule and gold electrodes. The cAMP molecules are smaller than the ATP, AMP, and ADP molecules because the phosphate group of cAMP is covalently bound to the hydroxyl group of ribose molecules. Therefore, there was considered to be a large difference in the initial state of attraction to the electrode. In summary, it was shown that using numerous features with machine learning made it possible to determine the slight differences in the molecules, which assisted in improving the identification accuracy of the single-molecule measurements.

## 5. Conclusions

A single-molecule quantum measurement method was used for cAMP and succeeded in detecting cAMP for the first time. Furthermore, this study succeeded in identifying signals from ATP, AMP, and ADP, which are molecules similar to cAMP. The single cAMP signal discrimination accuracy values for AMP, ADP, and ATP were found to be 0.82, 0.70, and 0.72, respectively. These values indicated a 99.9% accuracy when detecting more than ten signals. Furthermore, the features that contributed to the identification were identified. As a result that the features were affected by the interaction with the electrodes, the structural difference caused by the ribose of the phosphate site of cAMP was considered to be the reason for the differences compared to the ATP, ADP, and AMP signals. In this study, as a proof-of-concept, we demonstrated cAMP detection and its discrimination from other similar molecules. When multiple gap devices are developed and some kinds of cultured cell samples are mounted on the device, it is possible to visualize the cAMP related intercellular signaling pathway.

## Figures and Tables

**Figure 1 nanomaterials-11-00784-f001:**
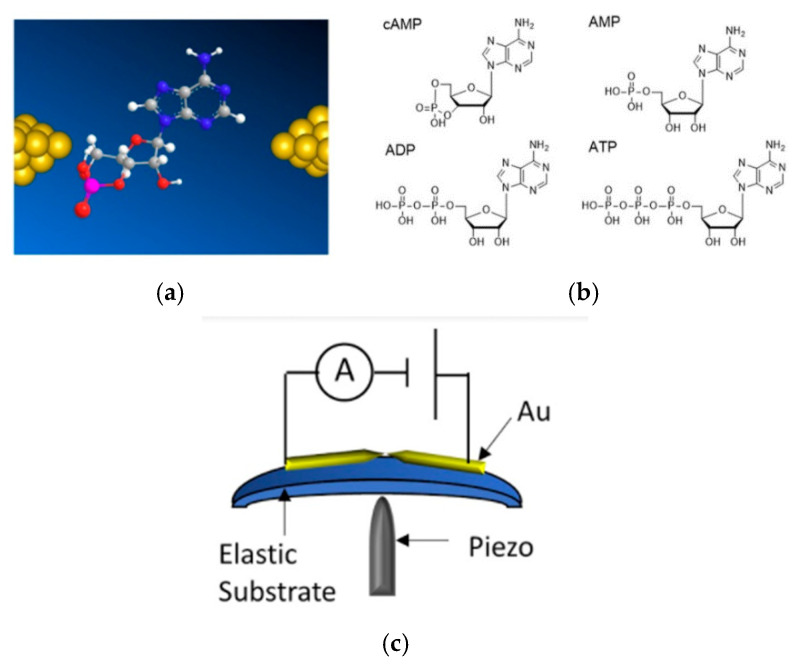
(**a**) Schematic image of single-molecule measurement of nucleotide-type second messenger, e.g., cyclic adenosine monophosphate (cAMP), using nanogap electrodes. (**b**) Chemical structures of four targeted molecules: cyclic adenosine monophosphate (cAMP), adenosine triphosphate (ATP), adenosine monophosphate (AMP), and adenosine diphosphate (ADP). (**c**) Device and measurement system. The mechanically controllable break-junction (MCBJ) system was used for the formation of nanogap electrodes.

**Figure 2 nanomaterials-11-00784-f002:**
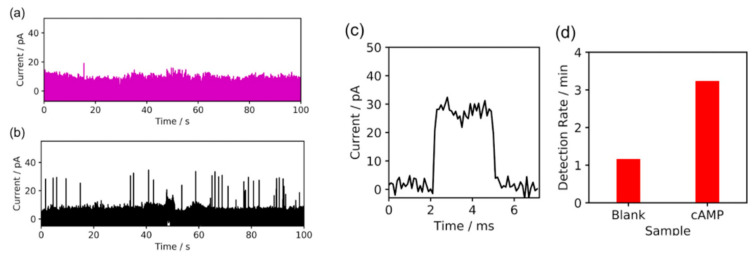
Current–time (*I–t*) profiles for signals obtained in water (**a**), and cAMP aqueous solution (**b**). (**c**) Enlargement of signal shown in (**b**) obtained in cAMP aqueous solution. (**d**) Comparison of signal frequency values for blank and cAMP, respectively. The signal frequencies are clearly different, with the signal frequency of the signals obtained in the cAMP aqueous solution (**right**: cAMP) significantly larger than that of the signals obtained in water (**left**: blank).

**Figure 3 nanomaterials-11-00784-f003:**
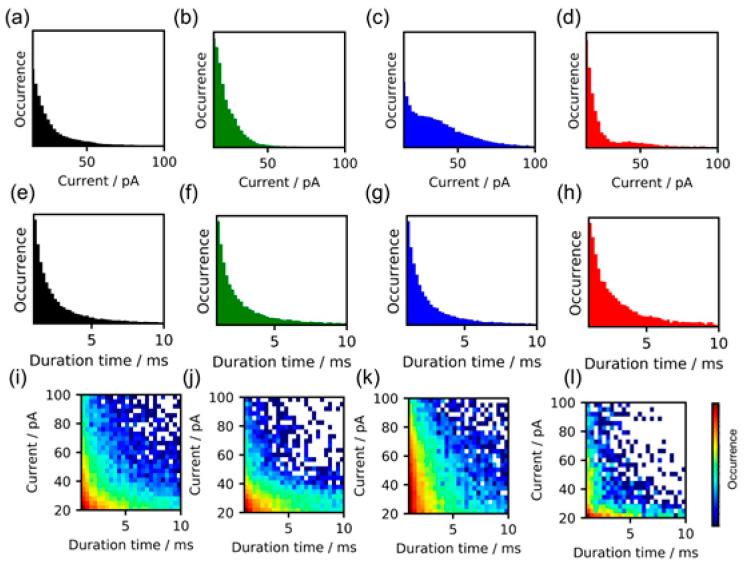
Statistical analysis results for obtained signals. In these analyses, at least 1000 signals were utilized. The occurrence in the histogram represents normalized signal numbers. (**a**–**d**) Current histograms at bias voltage of 100 mV for cAMP (**a**), AMP (**b**), ADP (**c**), and ATP (**d**). (**e**–**h**) Duration-time histograms of cAMP (**e**), AMP (**f**), ADP (**g**), and ATP (**h**). (**i**–**l**) Two-dimensional plots of *I_p_* and *t_d_* for cAMP (**i**), AMP (**j**), ADP (**k**), and ATP (**l**), and dependency of sample species.

**Figure 4 nanomaterials-11-00784-f004:**
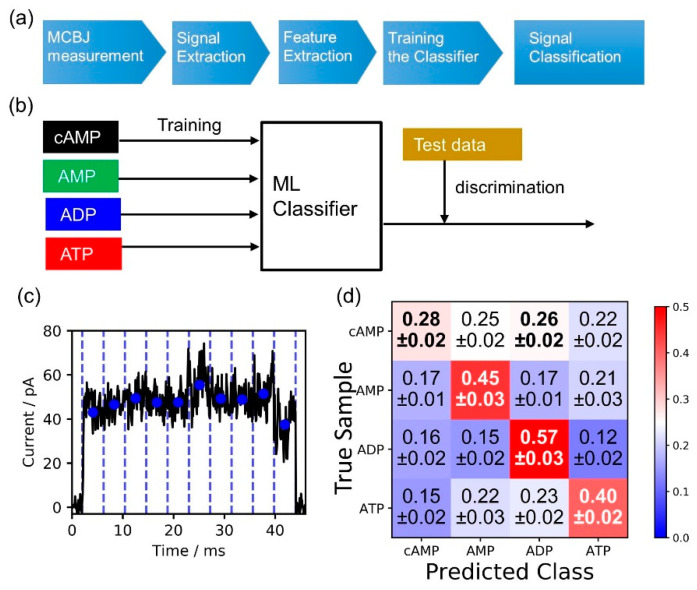
(**a**) Flow of data acquisition and analysis. (**b**) Scheme for machine learning analysis. (**c**) Definitions of 13 features parameter of signals used for machine learning analysis. These 13 feature parameters included the following: the peak value of the current signal (*I_p_*), average signal value (*I_ave_*), duration-time (*t_d_*), and ten-dimensional shape factors (*S_n_ = I_n_/I_p_ (n =* 1, 2, …, 10)). The average current value (blue circle) for each of the time regions (*I*_1_, *I*_2_, *I*_3_, *I*_4_, *I*_5_, *I*_6_, *I*_7_, *I*_8_, *I*_9_, and *I*_10_) was calculated. Each of the current values was normalized by the maximum current value of the signal (*I_p_*), and the values in each region were defined as shape factors (*S*_1_, *S*_2_, *S*_3_, *S*_4_, *S*_5_, *S*_6_, *S*_7_, *S*_8_, *S*_9_, and *S*_10_). (**d**) Evaluation results for discrimination accuracies of cAMP, ADP, AMP, and ATP. To calculate the confusion matrix for the signals, at least one thousand signals were obtained from the experiments and all the signals were used for the evaluation. The prediction ratios in the confusion matrix represent the numbers of signals predicted to be the molecules normalized by the total signals of the true molecules. The diagonal terms of the matrix indicate the discrimination accuracies of the molecules. The classifier was trained with 10,000 signals.

**Figure 5 nanomaterials-11-00784-f005:**
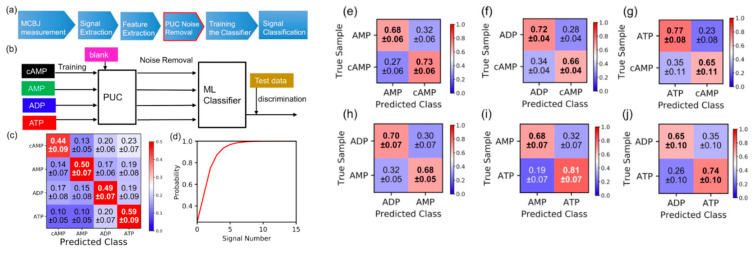
(**a**) Flow of data acquisition and analysis with positive and unlabeled data classification (PUC) method. In the PUC method, the signals obtained from the blank solution (control experiments) were used as positive signals. On the other hand, the signal data of the sample solution were regarded as unlabeled data because these signals were composed of both blank signals and the sample molecule signals. (**b**) Schematic of machine learning analysis using PUC method. (**c**) Evaluation results for discrimination accuracies of cAMP, ADP, AMP, and ATP using the PUC method. Compared to the discrimination accuracy value of 0.25 for cAMP without the PUC method, the average discrimination accuracy was also improved to 0.50. The classifier was trained with 1000 signals. (**d**) Relation between probability of accurate prediction by majority vote and number of signals. The accuracy was set to 0.5 for four classes. (**e**–**j**) Evaluation of discrimination accuracy of cAMP from other molecules, i.e., AMP, ATP, and ADP. The two-target discrimination accuracy values were found to be 0.70 for AMP/cAMP (**e**), 0.69 for ADP/cAMP (**f**), 0.75 for ATP/cAMP (**g**), 0.69 for AMP/ATP (**h**), 0.74 for ATP/AMP (**i**), and 0.69 for ADP/ATP (**j**).

**Figure 6 nanomaterials-11-00784-f006:**
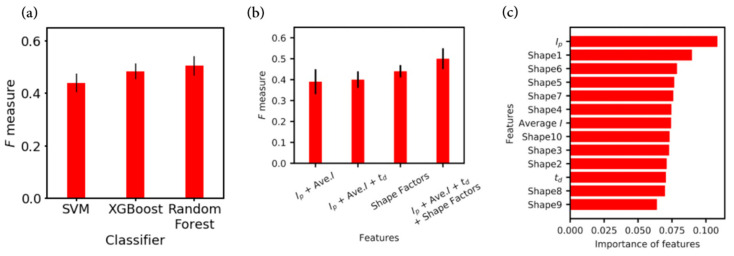
(**a**) Comparison of F-measure values for support vector machine (SVM), extreme gradient boosting (XGBoost), and random forest. The hyperparameters of SVM were determined by a grid search to have the best F-measure. All the algorithms succeeded in identifying with an F-measure of 0.42 or higher. Among several classification algorithms, including SVM and XGBoost, the random forest algorithm had the best discrimination accuracy. Its good discrimination accuracy indicated that all the prepared features (*I_p_*, *I_ave_*, *t_d_*, *S*_1_, *S*_2_, *S*_3_, *S*_4_, *S*_5_, *S*_6_, *S*_7_, *S*_8_, *S*_9_, and *S*_10_) used here were involved in the discrimination. An evaluation was made of the contributions to the discrimination of the different feature parameters. (**b**) Comparison of four sets of feature parameters, i.e., the maximum current value (*I_p_*), average current value (*I_ave_*), t_d_, and shape factors (*S_n_*). Among the four sets of the parameter combinations, the best combination was found to be maximum current value, average current value, and shape factors (*I_p_*, *I_ave_*, *t_d_*, *S_n_*). This confirmed that all the prepared features used here were important for the discrimination. In addition, among three sets of feature parameters, *I_p_*, t_d_, and the shape factors, the shape factors (*S_n_*) were found to provide the highest discrimination accuracy. This suggested that the shape factors were important. (**c**) Estimation of the importance of each feature parameter. The random forest classifier could evaluate the importance of each feature, which was an index of how much the discrimination accuracy decreased when each of the feature parameter was omitted. Among these indexes, two feature parameters, i.e., the maximum current value (*I_p_*) and first shape factor (S1), were found to be the most important.

## Data Availability

The data presented in this study are available on request from the corresponding authors.

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
