# Peer review of "Development of Single-Molecule Electrical Identification Method for Cyclic Adenosine Monophosphate Signaling Pathway"

_nanomaterials, 2021, doi:10.3390/nano11030784_

Round 1

Reviewer 1 Report

Yuki Komoto and coworkers report a single molecule detection system using a quantum device for a specific molecule of cyclic adenosine monophosphate.

It is hard to review from the scientific viewpoint because the description is very poor. For instance,

  1. Full terms should be shown at the first appearance of abbreviations, such as ATP and MCBJ.
  2. I cannot understand the description “Compared to the positive signals of Figure 2b, the averaged Ip for characteristic current signals are decreased from 33 pA for Figure 2a to 18pA for Figure 2b.” in the lines 146-148, at all. In the Figure 2a, the current values distribute between 10 pA and 20 pA: the average current may be close to 18 pA. On the other hand, the values of peak top at spike like current in Figure 2b distribute between 20 pA and 40 pA: the average may be 33 pA. This trend is not consistent with the description above at all.
  3. In the lines 148-150, Of what the signal behavior is in the “The signal behaviors are very similar to blank signals obtained in buffer aqueous solutions the previous reports [6].”? In addition, I think “in” is missing before “the previous reports”.

Please improve description including English with native speakers. I will be able to make scientific review after improvement of the manuscript.

Author Response

We would like to express many thanks to the referees for his or her valuable and helpful comments. We have studied the comments carefully and made the following revisions.

Reviewer 2 Report

Some comments

second vs secondary messenger, is not secondary preferred or mor common?

Define MCBJ

Are figure 1c and 1d (cAMP) consistent? with regard to detection rate?

In Fig 3 (i, j, k, and l) what do the different colors mean?

In Figure 4, how is 'discrimination accuracy' defined?

Line 295 what does 'signal conductivity' mean?

What exactly are Shape Factors, how are they defined?

Author Response

(The authors gave the same response as above.)

Reviewer 3 Report

In the article “Development of Single-molecule electrical identification 2 Towards cyclic AMP signaling pathway”, the authors investigate cyclic adenosine monophosphate (cAMP) and its related molecules with the single molecule quantum detection method. This method succeeded in detecting cAMP for the first time.

In my opinion, the subject of the paper is very interesting and deserves future developments. The proposed work appears very well structured and characterized. The experimental results are described and discussed thoroughly, and the significant and innovative aspects are highlighted.

I think that this work can become a valid reference for a broad range of researchers, dealing with the prominent, real-time single-molecule detection nanotechnologies and their potential applications within the life science fields.

Although I believe the work deserves to be published, however, the references need to be updated with recent papers published on the same topic.

Author Response

(The authors gave the same response as above.)

Round 2

Reviewer 1 Report

Yuki Komoto and coworkers report a single molecule detection system using a quantum device for a specific molecule of cyclic adenosine monophosphate. Although the manuscript is improved more or less, there are many concerns as follows.

  1. Are the shape factors defined as Sn = In/Ip (n = 1,2, …,10)? If so, the authors should clearly show the formulation and describe them in the smart way using n as suffix.
  2. Why can the feature parameter be used for discrimination? It should be due to the signal which is directly related to the electronic structure of molecules. Please discuss it or comment on it. Discussion at the lines 360 – 378 is not enough, because it is just qualitative but not quantitative.
  3. By comparing the Figures 3(a) and 3(b), and 3(e) and 3(f), these are quite similar to each other. I am wonder why these two types of signals can be recognized well. Even by machine learning, for recognition, data should be different. Please show the key feature of difference of these two sets of data.
  4. By comparing the Figures 3(e) – 3(g), the shapes of graph for < 5 ms are very very similar to each other. Especially, a peak at about 2 ms, followed by a dip, and 4 relatively high values and followed by a dip and a peak. I am afraid of artifact of the data handling.
  5. The authors claimed that “The results indicated that a larger number of noise signals were obtained in the cAMP signal measurements compared to those of the other molecules.” In the line 234-235. Is there any evidence or supporting data? If so, please show it clearly. No one can accept the claim, if the low accuracy is just attributed to high noise without any evidence. The measurement system should be the same, and then the signal to noise ratio should be the same in principle. Please show the quantitative data which show that noise levels of CAMP signal for supporting the authors’ hypothesis is higher than those of others.
  6. Relating to the previous comment, please show the improved the signal to noise ration after PUC noise removal process in comparison with that without PUC.
  7. The authors claimed that “For instance, when more than ten signals with a discrimination accuracy of 0.50 were obtained, a 99% accuracy is obtained (Figure 5d), which meets the criterion for practical use.” In the lines 266-268. I agree the authors’ claim if the sample contains only one type of molecule. However, I think discrimination of molecules in the mixed solution is required for application. In this case, this method cannot be applied. How to solve this problem?
  8. Please show the definition of “two-target discrimination accuracy value”.
  9. In addition to the comment 7, I am also curious about the application of this method. Will it be used in the cell? Maybe no. How to use this device for recognition of cAMP?
  10. There are still mechanical errors in the manuscript, e.g. lines 111 and 153.

From above reasons, the authors should check the data not to use incorrect data and revise the manuscript for further consideration of publication in nanomaterials.

Author Response

We would like to express many thanks to the referee for his or her valuable and helpful comments. We have studied the comments carefully and made the following revisions.
